# Age-Related Influence on Static and Dynamic Balance Abilities: An Inertial Measurement Unit-Based Evaluation [note 1]

**DOI:** 10.3390/s24217078

**Published:** 2024-11-03

**Authors:** Tzu-Tung Lin, Lin-Yen Cheng, Chien-Cheng Chen, Wei-Ren Pan, Yin-Keat Tan, Szu-Fu Chen, Fu-Cheng Wang

**Affiliations:** 1Department of Physical Medicine and Rehabilitation, Cheng Hsin General Hospital, Taipei 112, Taiwan; b101100094@tmu.edu.tw (T.-T.L.); a09830886@gmail.com (L.-Y.C.); braddom.chen@gmail.com (C.-C.C.); 2Graduate Institute of Gerontology and Health Care Management, Chang Gung University of Science and Technology, Taoyuan 333, Taiwan; 3Department of Mechanical Engineering, National Taiwan University, Taipei 106, Taiwanykeat11322@gmail.com (Y.-K.T.); 4Department of Physiology and Biophysics, National Defense Medical Center, Taipei 114, Taiwan

**Keywords:** balance, age-dependent changes, inertial measurement units, angular velocities, single-leg stance

## Abstract

Balance control, a complex sensorimotor skill, declines with age. Assessing balance is crucial for identifying fall risk and implementing interventions in the older population. This study aimed to measure age-dependent changes in static and dynamic balance using inertial measurement units in a clinical setting. This study included 82 healthy participants aged 20–85 years. For the dynamic balance test, participants stood on a horizontally swaying balance board. For the static balance test, they stood on one leg. Inertial measurement units attached to their bodies recorded kinematic data, with average absolute angular velocities assessing balance capabilities. In the dynamic test, the younger participants had smaller average absolute angular velocities in most body parts than those of the middle-aged and older groups, with no significant differences between the middle-aged and older groups. Conversely, in the single-leg stance tests, the young and middle-aged groups outperformed the older group, with no significant differences between the young and middle-aged groups. Thus, dynamic and static balance decline at different stages with age. These results highlight the complementary role of inertial measurement unit-based evaluation in understanding the effect of age on postural control mechanisms, offering valuable insights for tailoring rehabilitation protocols in clinical settings.

## 1. Introduction

Balance is essential for maintaining posture and responding to external perturbations [1]. It plays a crucial role in daily activities such as walking, standing, and reaching. Impaired balance is associated with falls among older individuals [2,3,4]. Falls can lead to severe health problems, including fractures, reduced mobility, hospitalization, and even death [5,6]. Therefore, assessing balance ability is of utmost importance.

Poor balance is often observed in older adults and is a risk factor for falls [7]. An estimated 13% of adults aged 65–69 years of age experience imbalance, increasing to 46% in those aged ≥85 years [8]. Age-related decline in postural stability can result from impairments in the sensory, motor, and central processing systems. Osoba et al. reported that older adults experience an age-related decline in the somatosensory and vestibular systems along with a reduced ability to adapt to environmental changes and maintain balance [9]. Another critical factor is the motor system. Muscle strength reaches its peak by the age of 30 years in men and is maintained until approximately 50 years of age. However, after this age, there is a gradual decline in strength, estimated to be approximately 12–15% per decade until the eighth decade of life [10]. In addition to the age-related decline in sensory input, there is evidence of changes in central processing mechanisms [11]. These changes may impair the integration of sensory information and reduce the ability to compensate for unreliable or conflicting sensory inputs.

Traditional balance measures include the Timed Up and Go Test (TUG) [12], Functional Reach Test [13], Berg Balance Scale [14], Fullerton Advanced Balance Scale [15], and Dynamic Gait Index [16]. However, these measures are often subjective and provide only categorical results. Additionally, the ceiling effect and poor early detection capabilities limit the utility of conventional balance scales. Recent studies have proposed more objective assessments using instrumented and quantitative balance measures. Quantitative parameters provide more precise data than categorical parameters. For instance, Prieto et al. applied force plates to measure the center of pressure for balance assessment [17]. Lim et al. employed Microsoft’s Kinect to measure the center of body mass to evaluate balance ability [18]. Martin et al. utilized an optical motion capture system and surface electromyography to assess kinematic data and muscle activation in patients with multiple sclerosis [19]. However, these measurement methods are often limited by experimental location and cost. In contrast, wearable inertial measurement units (IMUs) are frequently used to measure kinematic data during experiments. For example, Liu et al. [20] applied IMUs to measure the gait performance and balance ability of Latin dancers and concluded that dance training improved their balance ability. Lin et al. [21] used IMUs to evaluate the postural stability of yoga instructors and found that yoga practice improved postural stability. Tien et al. employed IMUs to analyze gait features in patients with Parkinson’s disease [22].

Although numerous studies have utilized inertial measurement units (IMUs) to assess gait and postural stability, few have specifically examined age-related changes in balance [23,24,25,26]. Most of these studies focused solely on either static or dynamic balance, with IMUs mainly positioned on the lower trunk to assess trunk stability [23,25,26], and many limited their analyses to accelerations [23,24]. Park et al. [24] analyzed the effect of age on both static balance and gait, using multiple measures to evaluate postural sway, step initiation, gait, and turning; however, all measures were derived solely from acceleration data. Marchesi et al. [23] used a robotic platform to investigate reactive postural components with both eyes open and eyes closed in participants aged 20 to 90. Nonetheless, balance performance was quantified using trunk movements measured solely by an accelerometer on the sternum, which may not capture the full range of postural adjustments across the body. Similarly, O’Brien et al. [26] employed a single lower back IMU to quantify gait features during the 10 m walk test and Timed Up and Go Test in participants ranging from 20 to 70 years of age.

Using a single IMU limits kinematic data to one body segment, which restricts the assessment of coordination between different body parts. This approach may miss crucial segmental movements needed for balance, as effective balance often involves coordinated actions across multiple joints. It can also reduce sensitivity to subtle balance impairments, especially in those with age-related or neurological conditions. Additionally, assuming that trunk movement represents overall body balance can be inaccurate, particularly during dynamic activities. Incorporating data from other body parts, such as the hips or feet, could provide a more comprehensive analysis. Moreover, evaluating only static or dynamic balance abilities in age-related balance decline can provide a limited understanding of the underlying mechanisms and risk factors for falls.

Balance ability is typically divided into static and dynamic aspects. Static balance refers to the capacity to remain upright and maintain the center of gravity within the base of support while standing still [27]. In contrast, dynamic balance refers to the ability to maintain stability while shifting weight, often while altering one’s base of support [28]. Single-leg stance and balance board are commonly used balance tests for evaluating static balance and dynamic balance, respectively. Previous studies investigating aged-related balance decline during one-leg stance have often relied on traditional methods, such as measuring stance time or using force plates [29,30]. Similarly, research on the age-related deterioration of dynamic balance using balance boards has primarily utilized reaction time measurements or force plates [31,32,33]. The above approach primarily focused on overall stability metrics derived from a central point of measurement and may fail to capture the intricate contributions of various body segments to postural control. A notable research gap exists in the use of more advanced methodologies, such as deploying multiple IMUs across different body segments, which could yield a more detailed and comprehensive analysis of balance dynamics. Furthermore, these studies have often not included a sufficiently diverse range of age groups, typically concentrating on either young or older adults while overlooking middle-aged individuals [31,32]. Rubens et al. and Parreira et al. assessed age-related differences during one-leg stance in only young and older adults [29,30]. This results in a limited understanding of how balance evolves throughout the aging process. The application of advanced technology and a more inclusive study population may provide deeper insights into age-related changes in balance ability.

To address these research gaps, we used a larger number of IMUs from various body parts, including the waist as the primary outcome measure and other body segments, including the chest, knees, ankles, and feet, to assess age-related changes in both static and dynamic balance. Two balance indices—average absolute angular velocity ($J_{\omega }$) and average absolute linear acceleration ($J_{a}$)—were employed. Based on the prior literature regarding balance and aging [2,3,4,7,9], the primary objective of this study is to determine whether there are significant age-related differences in balance indices among young, middle-aged, and older adults during both static and dynamic balance tasks. We hypothesize that balance indices will significantly differ among young, middle-aged, and older adults, with an age-related decline in stability observed in both static (single-leg stance) and dynamic (balance board) balance contexts. A secondary objective is to explore whether there is a significant correlation between age and balance indices within both static and dynamic contexts, as age has been linked to changes in postural stability due to sensory and motor system decline [9,10]. To address these hypotheses, we specifically address the following research questions: Are there statistically significant differences in balance indices among young, middle-aged, and older adults during the single-leg stance test? Are there statistically significant differences in balance indices among young, middle-aged, and older adults during the balance board test? Is there a significant correlation between age and balance indices in the single-leg stance test? Is there a significant correlation between age and balance indices in the balance board test?

## 2. Materials and Methods

### 2.1. Study Design

This cross-sectional study was approved by the Institutional Review Board of Cheng Hsin General Hospital, Taiwan ((1026)112-17). All balance performance data were collected from the participants who provided written informed consent with permission for the use of all photographs included in this manuscript.

### 2.2. Participants

We recruited 82 participants for the balance tests: 21 young adults (aged 18–29 years), 29 middle-aged adults (aged 30–64 years), and 32 older adults (≥65 years). The selection criteria for the participants were as follows: (1) good overall health, (2) no history of surgery on the lower limbs, (3) absence of active musculoskeletal injuries or neurological conditions affecting walking ability, and (4) ability to walk independently for more than 50 m without assistance.

### 2.3. Inertial Measurement Unit System

We used the Opal IMU system (APDM Wearable Technologies, Portland, OR, USA) [34] with wearable IMUs to measure the kinematic data of the participants during the experiments.

### 2.4. Dynamic Balance: Balance Board Stance Test

Many different unstable boards have been utilized to assess dynamic standing balance [35,36]. Wobble boards are frequently used for conventional dynamic balance assessment [37,38,39]. Computerized Dynamic Posturography (CDP) is also a widely accepted method for assessing dynamic standing balance abilities [40,41]. However, a wobble board setup usually lacks handrails, making it more challenging for older adults who are encountering it for the first time, raising some safety concerns. Additionally, CDP is often more expensive and limited by its experimental setting. Therefore, we decided to assess dynamic standing balance using a balance board, the Pro Balancer (Monitored Rehab Systems, Haarlem, Netherland), commonly used for balance training in clinical settings for patients with stroke and spinal cord injuries.

The balance board, as depicted in Figure 1a, has a dimension of 60 × 35 cm with handrails surrounding the participant’s anterior and left and right sides. The board was secured to a frame with four steel cables, allowing horizontal movement and the possibility of self-induced perturbations. The steel cables suspended the balance board, and the lengths of the cables had no notable variation during the tests. Therefore, we can assume that the stiffness of these cables is high and regard the system as a pendulum with simple harmonic motions. The participants try to balance the system by body movements through different senses, such as vision and touch. Hence, we can estimate the participants’ balance ability by their body movements when balancing the system.

The participants stood on the board with their arms crossed in front of their chests to minimize potential compensatory strategies to maintain balance. Initially, the participants maintained their balance with their feet separated at a distance approximately equal to the width of their shoulders for 60 s, as illustrated in Figure 1b. Subsequently, they maintained balance with their feet together for another 60 s, as shown in Figure 1c. The participants were permitted to rest between tests, if necessary, and terminate the tests any time by holding onto the rack.

During the balance board stance test, seven IMUs were attached to the participants: one on the waist, one on the chest, two on the knees, two on the ankles, and one on the right foot. The IMUs recorded the kinematic data, including 3-axial accelerations and 3-axial angular velocities, at a sampling rate of 128 Hz. Data collection was immediately halted in the event of loss of balance. If a loss of balance occurred within the 60 s period, the trial was repeated twice, and the longest successful trial was selected for the analysis.

### 2.5. Static Balance: Single-Leg Stance Test

A one-leg stance test was conducted to evaluate the static postural stability of the participants. During the test, they were instructed to stand on their right foot for 60 s while lifting their left leg and then repeat the test by standing on their left foot for 60 s with their right leg lifted. Eight IMUs were attached to each participant: one on the waist, one on the chest, two on the wrists, two on the ankles, one on the standing foot, and one on the head. The IMUs recorded the kinematic data throughout the test.

During the tests, the participants crossed their arms in front of their chest, as shown in Figure 2, to minimize compensatory balance strategies. Participants were allowed to rest between tests if necessary and could terminate the test at any time by lowering their legs. Data collection was immediately halted in the event of loss of balance. If balance loss occurred within the 60 s duration, the trial was repeated twice, and the longest successful trial was selected for the analysis.

### 2.6. Evaluation of Dynamic Balance and Static Balance

We defined the following two performance indices to quantify the participant’s balance ability:(1)$$J_{\omega }^{k}=\frac{1}{N}\sum_{i=1}^{N}\left|\omega^{k}\left(i\right)\right|=\frac{1}{N}\sum_{i=1}^{N}{\left({\left(\omega_{x}^{k}\left(i\right)\right)}^{2}+{\left(\omega_{y}^{k}\left(i\right)\right)}^{2}+{\left(\omega_{z}^{k}\left(i\right)\right)}^{2}\right)}^{1/2}\;\mathrm{and}$$
(2)$$J_{a}^{k}=\frac{1}{N}\sum_{i=1}^{N}\left|a^{k}\left(i\right)\right|=\frac{1}{N}\sum_{i=1}^{N}{\left({\left(a_{x}^{k}\left(i\right)\right)}^{2}+{\left(a_{y}^{k}\left(i\right)\right)}^{2}+{\left(a_{z}^{k}\left(i\right)\right)}^{2}\right)}^{1/2},$$
where $k\in \left\{waist,\;chest,\;wrist,knee,\;ankle,\;foot,head\right\}$ indicates the angular velocities and linear acceleration of the chest, waist, wrist, knee, ankle, foot, and head. *N* is the total number of samples. $\omega_{x}^{k}\left(i\right),\;\omega_{y}^{k}\left(i\right),\;\mathrm{and}\;\omega_{z}^{k}\left(i\right)$ and $a_{x}^{k}\left(i\right),\;a_{y}^{k}\left(i\right),\;\mathrm{and}\;a_{z}^{k}\left(i\right)$ are the 3-axial angular velocities and accelerations, respectively, from the IMU in the *i*-th sample; in other words, $\omega^{k}\left(i\right)$ and $a^{k}\left(i\right)$ represent the absolute angular velocity and linear acceleration, respectively, at the *k* location in the *i*-th sample. $J_{\omega }^{k}$ represents the average absolute angular velocity. $J_{a}^{k}$ represents the average absolute linear acceleration. The participant’s balance ability was better with a smaller J (i.e., the participant did not need much effort to maintain balance).

### 2.7. Statistical Analysis

The Shapiro–Wilk test was employed to evaluate data distribution. If the data followed a normal distribution, results were reported as mean ± standard deviation (SD); otherwise, the median with interquartile range (IQR) was used as a more appropriate measure. To determine the overall difference between the groups, we used either a one-way ANOVA with a post hoc pairwise comparison of the Tukey–Kramer test for normally distributed data or the Kruskal–Wallis H test with a post hoc pairwise comparison of Dunn’s test with Bonferroni correction for non-normally distributed data. Pearson’s correlation was conducted for normally distributed data, while Spearman’s correlation was used for non-normally distributed data to examine associations between $J_{\omega }$ and age. The correlation was categorized as “very strong” if the correlation coefficient (r) was ≥0.80, “strong” if r was ≥0.6 but <0.80, “moderate” if r was ≥0.4 but <0.6, “weak” if r was ≥0.20 but <0.40, and “very weak” if r was <0.20 [42]. Microsoft Office Excel 2019 software and SPSS statistical software 22.0 were used to analyze the IMU data. Statistical significance was set at *p* < 0.05.

## 3. Results

### 3.1. Demographic Data

A total of 82 participants were recruited and categorized into three age groups: 21 young adults (mean age ± SD [range]: 25 ± 2.43 years [21–29 years]), 29 middle-aged adults (42.66 ± 8.94 years [30–59 years]), and 32 older adults (72.13 ± 3.92 years [66–84 years]). Further details on the participants’ demographics are presented in Table 1. No statistically significant differences were detected among the three groups in terms of sex distribution (*p* = 0.902) or height (*p* = 0.280). However, weight, body mass index (BMI), and TUG test results differed significantly among the groups (*p* = 0.050, *p* = 0.009, and *p* < 0.001, respectively). Post hoc analysis showed that middle-aged and older adults had a significantly higher BMI than young adults (BMI: young vs. middle-aged: *p* = 0.017, young vs. older: *p* = 0.019, middle-aged vs. older: *p* = 0.993). Additionally, the older adult group exhibited longer TUG test durations than the young and middle-aged groups (young vs. middle-aged: *p* = 0.648, young vs. older: *p* < 0.001, middle-aged vs. older: *p* < 0.001). Notably, only older adults reported a history of falls.

### 3.2. Evaluation of Dynamic Balance: Balance Board Stance

We first evaluated the dynamic balance ability using a balance board stance test. Kinematic data from the IMUs were analyzed to compare the angular velocities across various body segments. We used $J_{\omega }$ and $J_{a}$ at the waist as the primary outcome measure as in previous studies and also analyzed different body segments, including the chest, knees, ankles, wrists, and feet, across the young, middle-aged, and older adult groups during both the feet-apart and feet-together stances. The analyses of $J_{\omega }$ are presented in Table 2 and Table 3. The results of $J_{a}$ are shown in Appendix A. There were no significant differences in $J_{a}$ among young, middle-aged, and older adults at the waist. Therefore, we focused on $J_{\omega }$ as our main outcome measure.

During the feet-apart stance, middle-aged and older adults exhibited significantly higher $J_{\omega }$ across all body segments than young adults (Table 2). However, there were no significant differences between middle-aged and older adults among all body segments.

During the feet-together stance, middle-aged and older adults exhibited significantly higher $J_{\omega }$ across all body segments than young adults, except for the right ankle and foot (Table 3). Similarly, no significant differences in average absolute angular velocities were observed between the middle-aged and older adult groups for any body segment during the feet-together stances.

We investigated the relationship between age and balance indices in different body segments. During the feet-apart stance, Spearman’s correlation coefficients revealed a moderate positive correlation between age and $J_{\omega }$ at the waist, chest, right knee, left knee, right ankle, left ankle, and foot (waist: *r* = 0.41, chest: *r* = 0.50, right knee: *r* = 0.49, left knee: *r* = 0.45, right ankle: *r* = 0.41, left ankle: *r* = 0.45, foot: *r* = 0.49; all *p* < 0.0001; Figure 3). Concurrently, moderate and positive correlations were observed between age and $J_{\omega }$ at the waist, chest, right knee, left knee, right ankle, left ankle, and foot (waist: *r* = 0.51, chest: *r* = 0.59, right knee: *r* = 0.47, left knee: *r* = 0.44, right ankle: *r* = 0.44, left ankle: *r* = 0.44, foot: *r* = 0.46; all *p* < 0.0001; Figure 4) during the feet-together stance. These findings demonstrate an age-related decline in dynamic balance abilities.

### 3.3. Evaluation of Postural Stability: Single-Leg Stance Test

We also deployed IMUs on various body segments of the participants to assess their static balance ability using kinematic IMU data during the one-leg stance tests. Analyses of $J_{\omega }$ are presented in Table 4 and Table 5. The results of $J_{a}$ are shown in Appendix A. There were no significant differences in $J_{a}$ among young, middle-aged, and older adults at the waist. Therefore, we focused on $J_{\omega }$ as our main outcome measure.

Both young and middle-aged adults demonstrated lower average absolute angular velocities in comparison with older adults during one-leg stance tests.

These findings underscore the reduced static balance ability of older adults compared with both young and middle-aged adults.

Notably, significant differences between the young and middle-aged groups were observed in the post hoc analysis for only four out of eight body segments, including the chest, right and left wrists, and right foot, during the right foot stance (Table 4). Similarly, significant differences were found in the post hoc analysis for only two out of eight body segments, specifically the left ankle and left foot, during the left foot stance (Table 5).

We further investigated the relationship between age and balance indices across various body segments during one-leg stance tests. In the right foot stance, Spearman’s correlation coefficients indicated a strong positive correlation between age and $J_{\omega }$ at the waist, chest, right wrist, left wrist, left ankle, left foot, and head (waist: *r* = 0.72, chest: *r* = 0.68, right wrist: *r* = 0.74, left wrist: *r* = 0.73, left ankle: *r* = 0.64, left foot: *r* = 0.74, head: *r* = 0.71; all *p* < 0.0001; Figure 5) and a moderate correlation in the right ankle (*r* = 0.43; *p* < 0.0001; Figure 5).

Simultaneously, a strong positive correlation was observed between age and $J_{\omega }$ at the waist, chest, right wrist, left wrist, left ankle, foot, and head (waist: *r* = 0.67, chest: *r* = 0.68, right wrist: *r* = 0.72, left wrist: *r* = 0.72, left ankle: *r* = 0.61, right foot: *r* = 0.76, head: *r* = 0.70; all *p* < 0.0001; Figure 6) during the left foot stance. Moreover, there was a moderate correlation between age and $J_{\omega }$ in the right ankle (*r* = 0.58, *p* < 0.0001; Figure 6). These results highlight an age-associated deterioration in static balance capabilities.

## 4. Discussion

In this study, we measured age-dependent changes in static and dynamic balance using IMUs in a clinical setting. In the dynamic test assessed employing the balance board test, young participants had smaller average absolute angular velocities in most body parts than the middle-aged and older groups, with no significant differences between middle-aged and older groups. In contrast, in the single-leg stance tests for evaluating static balance abilities, the young and middle-aged groups outperformed the older group. Our findings indicate that dynamic and static balance decline at different stages with age. Based on these results, we can infer that dynamic balance (balance board) declines more rapidly than static balance (single-leg stance). To our knowledge, this observation has not been highlighted in previous studies.

In this study, the dynamic balance ability was assessed using a balance board stance test. The results indicated a higher balance index in the feet-together stance than that in the feet-apart stance across all body locations and age groups, suggesting greater postural sway in the more challenging, narrow stance. This demonstrates the ability of IMUs to differentiate between tests with varying difficulty levels. Additionally, in both feet-apart and feet-together stances, middle-aged and older adults exhibited significantly higher average absolute angular velocities across all body segments than young adults. However, there was no significant difference between the middle-aged and older adults. This suggests that a greater increase in postural sway occurs when transitioning from young adulthood to middle age, with a smaller increase observed as one progresses from middle to older age. This may be attributed to the earlier deterioration of the balance ability required for this test, which stabilizes after middle age. According to Horak et al. [43], six components are essential for postural stability: biomechanical constraints (functional stability limits, strength, and static stability), movement strategies, sensory strategies, orientation in space, control of dynamics, and cognitive processing. The balance board stance test employed in this study included components of the underlying motor system, static stability, reactive postural control, and sensory integration [44]. Thus, an age-related decline in the somatosensory and vestibular systems, along with a reduced ability to adapt to environmental changes, may contribute to the decline in dynamic balance ability beginning in middle age. However, further research is required to understand the relative contribution of each component.

The single-leg stance is a widely used clinical tool for evaluating balance in individuals with various balance disorders [45]. In contrast to the findings from the balance board stance test, the single-leg stance test showed significant differences between young and older adults, as well as between middle-aged and older adults, but no significant difference was observed between the young and middle-aged groups. This suggests that increased postural sway is more prominent when transitioning from middle age to older age compared to the transition from young adulthood to middle age during a single-leg stance. Our results align with those of a previous study that evaluated postural stability through sway velocity, demonstrating significantly decreased stability in the one-leg stance with eyes open by the sixth decade of life [31]. According to Sibley et al. [44], the single-leg stance test incorporates balance components related to the underlying motor systems and static stability. Muscle strength tends to decline more rapidly after 65 years of age [46]. Muscle strength reaches its peak by the age of 30 years in men and maintains this level until approximately 50 years of age. However, after this point, a gradual decline occurs, estimated at approximately 12–15% per decade, until individuals reach their eighth decade of life [10,47]. Therefore, a decline in muscle strength among older adults may have contributed to their poorer performance in the single-leg stance test. For example, weakness in the hip abductor muscles may result in slower left–right weight shifting, and weakness in the ankle muscles may increase the anterior–posterior postural sway.

The present study demonstrated age-related changes in both static and dynamic balance abilities, consistent with previous findings. Previous studies investigating the influence of age on postural control have mainly employed a single inertial IMU to assess balance across various age groups [23,25,26]. While this approach streamlines data collection, it is associated with several limitations. Firstly, the use of a single IMU restricts kinematic data to a specific body segment, hindering a comprehensive evaluation of intersegmental coordination and interaction. Secondly, this approach may overlook essential segmental movements involved in balance maintenance, as effective balance often requires coordinated actions among multiple joints [48]. Thirdly, relying on a single IMU may diminish sensitivity in detecting subtle balance impairments, particularly in populations with age-related or neurological conditions [49]. Moreover, the assumption that trunk movement is representative of overall body balance may not hold true during dynamic activities. Finally, the placement of a single IMU can introduce measurement bias, as results may not accurately generalize if the selected location does not fully capture the employed balance control strategies. To address the limitations of previous studies, we employed a larger number of IMUs to measure kinematic data across various body segments, including the chest, waist, knees, ankles, and feet. Our results revealed significant differences between the young and middle-aged groups in only three of the six body segments: the chest, wrist, and foot during the single leg-stance test. These findings highlight the importance of considering multiple body segments and balance assessment methods when investigating age-related changes in postural control. While the waist may be a valuable marker for overall balance, additional measurements from other body parts may provide more nuanced insights into the specific mechanisms underlying balance decline with age. Further research is needed to explore the implications of these findings for clinical practice and intervention strategies. Understanding the specific body segments and balance assessment methods that are most sensitive to age-related changes can inform the development of targeted interventions to improve balance and reduce the risk of falls in older adults.

In contrast to previous studies that relied solely on parameters calculated from linear accelerations to assess balance and gait [23,24], we applied IMUs to measure both the participants’ accelerations and angular velocities during the tests, addressing the limitations associated with using parameters derived from linear acceleration data compared to angular velocity. Our recorded data indicated that the linear acceleration measurements were susceptible to the installation position of the IMU. For example, variations in distance can significantly affect the magnitude of the measured acceleration, while changes in the installation angle can influence the projection of the acceleration components. Conversely, angular velocity measurements are more robust against variations in the IMU installation position, making them less error-prone and more reliable. Therefore, this study evaluated participants’ balance control ability using the average absolute angular velocity. Also, acceleration data primarily capture linear movements, potentially limiting sensitivity to the rotational aspects of balance. This focus could result in overlooking subtle rotations around joints, such as those at the ankle and hip, that are essential for maintaining postural control. Angular velocities, on the other hand, allow for a more detailed differentiation of movement phases, facilitating nuanced analyses of dynamics like forward leans versus corrective motions during balance recovery [25,50]. Furthermore, angular velocity measurements are more adept at capturing joint-specific contributions to balance, providing insights into the coordinated rotations of the ankle, knee, and hip, while acceleration data may not convey these details as clearly. Interpreting complex movements can also be more challenging with acceleration data as they may primarily reflect changes in position rather than the critical rotational adjustments needed for maintaining balance. In clinical settings, angular velocities often show a more direct correlation with the assessment of movement disorders, such as rigidity or bradykinesia in conditions like Parkinson’s disease, which may enhance their relevance [51]. Moreover, we previously defined a balance index called average absolute angular velocity, which successfully differentiated the balance abilities between Latin dancers, yoga instructors, and age-matched adults [20,21]. Overall, the integration of multiple IMUs and the use of both angular velocity and linear acceleration can provide a more comprehensive and nuanced understanding of balance dynamics, particularly in studies investigating the influence of age on postural control.

Combining simple clinical balance tests with IMUs offers a more objective and straightforward approach to evaluating balance ability. Wearable IMUs are small, inexpensive, and easy to implement, making it feasible for assessing balance in real-life settings. In hospital environments, IMUs can be utilized to quickly evaluate balance in outpatient departments. Prior to this study, our team also assessed the balance ability of Latin dancers [20] and yoga instructors [21] using IMUs and the average absolute angular velocity index and found significantly better balance performance in both groups. Building on these successful experiences, we plan to conduct further research in different populations using the same method. IMUs can also be employed to investigate various aspects of balance ability, potentially leading to more accurate fall risk predictions.

Our study has certain limitations. First, it was conducted with a relatively small sample size. Second, we only included healthy participants and excluded those with musculoskeletal injuries or neurological conditions. Participants with a history of multiple falls were also excluded owing to safety concerns. Future studies should include participants with diverse health conditions, ensuring safety with additional precautions and equipment, to better identify the characteristics of high-risk populations. Third, there were significant differences in body weight and BMI across different age groups. However, as weight and BMI generally increase with age [52], our demographic data may accurately reflect real-world trends. Finally, an ideal dynamic balance evaluation should involve changes in the base of support such as during walking or performing the TUG test. Therefore, we plan to incorporate dynamic tests into future assessments.

## 5. Conclusions

This study investigated the age-related changes in static and dynamic balance using IMUs in a clinical setting. In the balance board test, which evaluates dynamic balance abilities, young participants exhibited smaller average absolute angular velocities across most body parts than the middle-aged and older participants. However, no significant differences were found between the middle-aged and older groups. Conversely, in the single-leg stance tests for evaluating static balance abilities, the young and middle-aged groups outperformed the older group, with no significant differences between the young and middle-aged groups. Our findings suggest that dynamic and static balance decline at different stages of aging. However, further studies with different tests or populations are needed for a more comprehensive evaluation and identification of high-risk groups at an early stage, which could help prevent potential falls. In the future, we may develop tailored strategies for balance training based on kinematic data from balance tests and monitor the effects of these interventions.

## Figures and Tables

**Figure 1 sensors-24-07078-f001:**
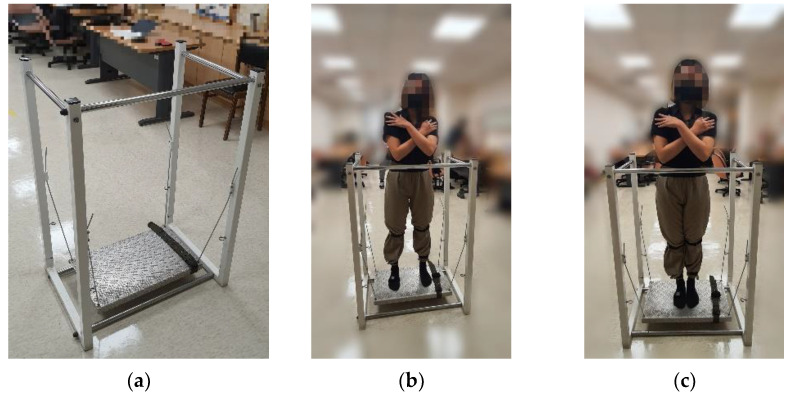
The balance board stance tests. Representative images of (**a**) the balance board; (**b**) the feet-apart stance; and (**c**) the feet-together stance.

**Figure 2 sensors-24-07078-f002:**
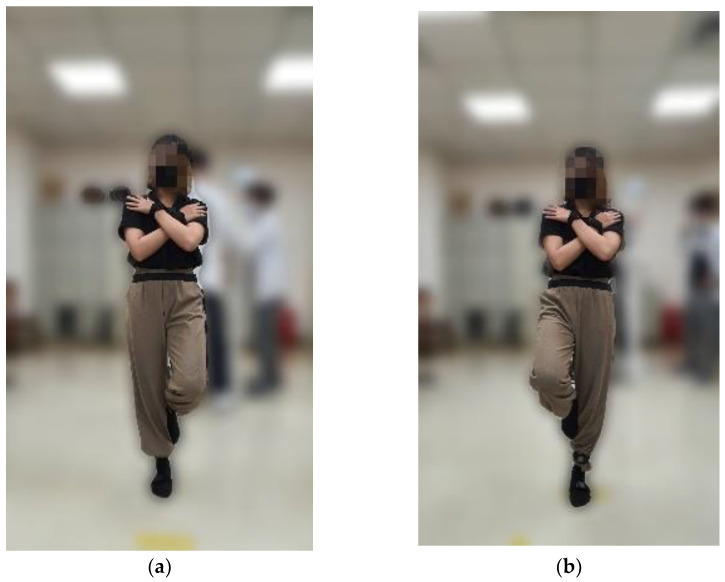
Single-leg stance test. Representative images of (**a**) right leg standing and (**b**) left leg standing.

**Figure 3 sensors-24-07078-f003:**
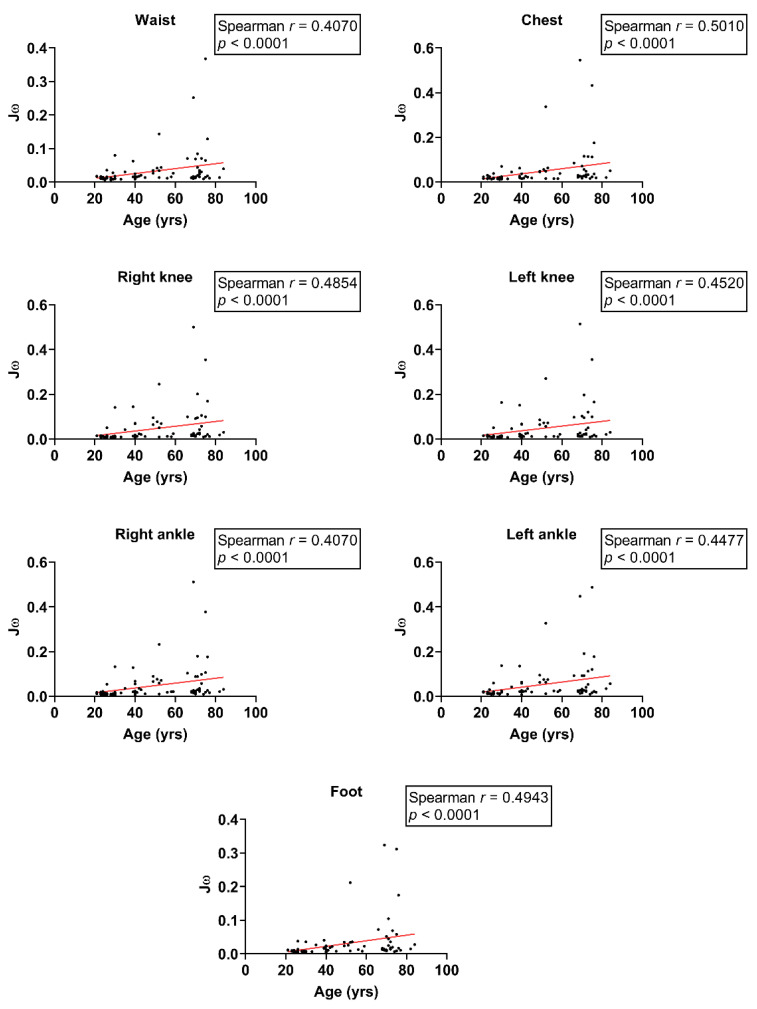
Scatter plot of $J_{\omega }$ and age with feet-apart stance on the balance board.

**Figure 4 sensors-24-07078-f004:**
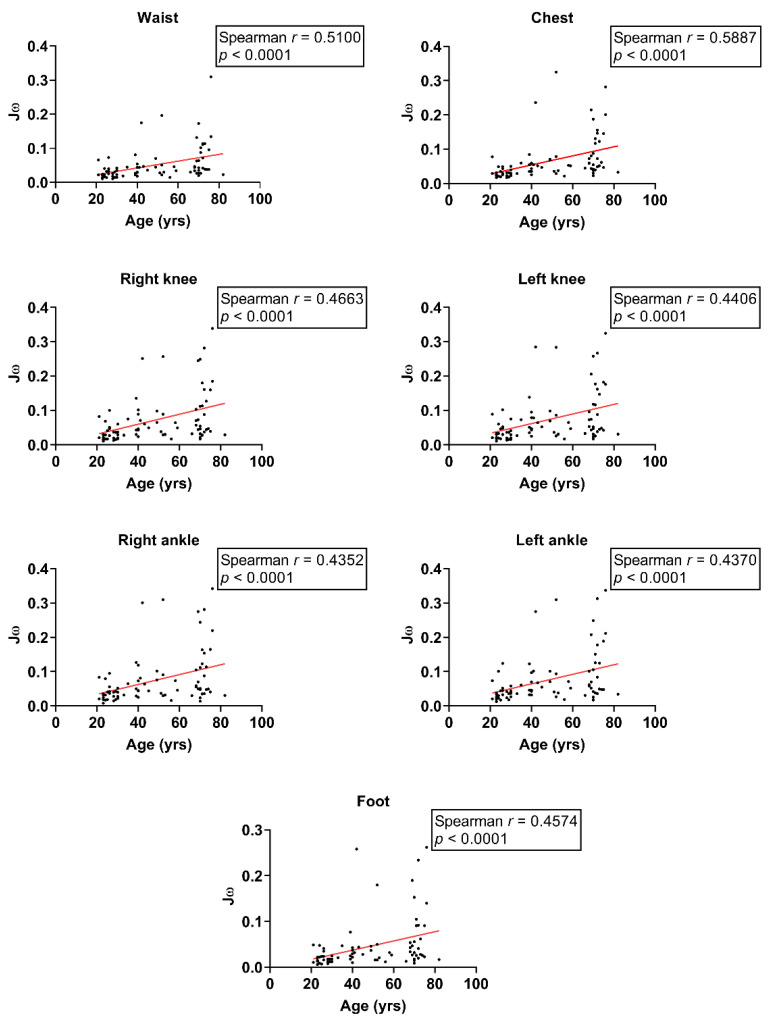
Scatter plot of $J_{\omega }$ and age with feet-together stance on the balance board.

**Figure 5 sensors-24-07078-f005:**
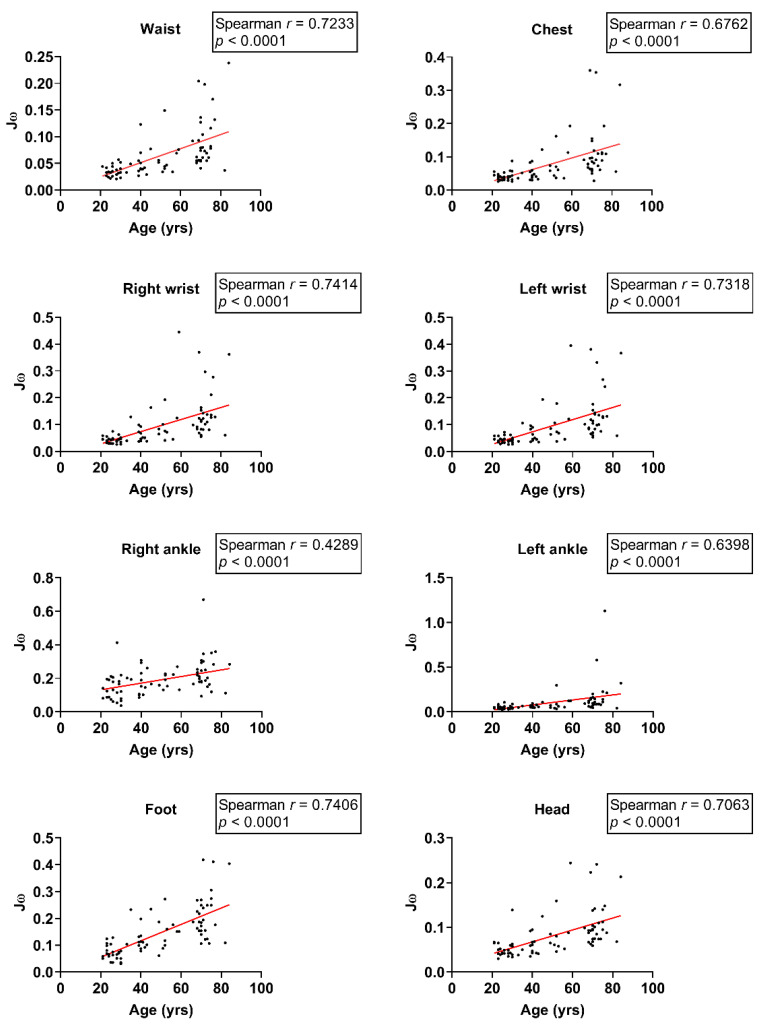
Scatter plot of $J_{\omega }$ and age during single-leg stance on the right foot.

**Figure 6 sensors-24-07078-f006:**
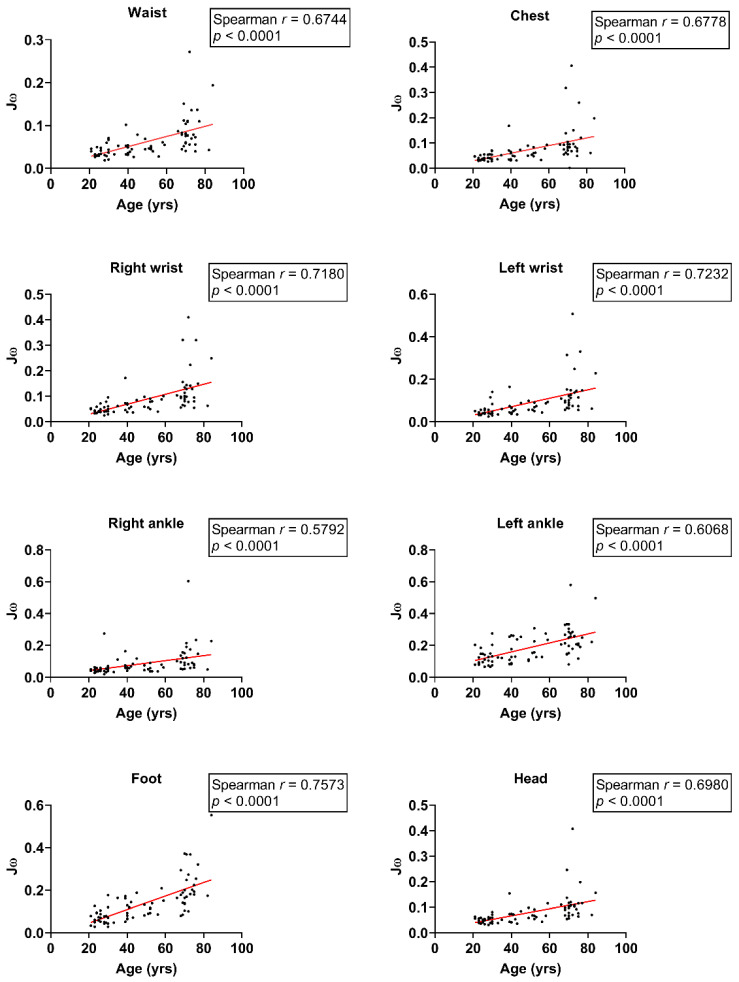
Scatter plot of $J_{\omega }$ and age during single-leg stance on the left foot.

**Table 1 sensors-24-07078-t001:** Demographic data.

Characteristics	Young	Middle-Aged	Older	*p*-Value
Age, years	25 ± 2.43	42.66 ± 8.94	72.13 ± 3.92	-
Female, *n* (%)	10 (47.6%)	14 (48.3%)	17 (53.1%)	0.902
Weight, kg	65.25 ± 11.01	68.90 ± 15.63	59.48 ± 12.61	0.050 *
Height, cm	164.86 ± 8.22	166.86 ± 8.76	163.22 ± 9.28	0.280
Body mass index	21.76 ± 3.58	24.50 ± 3.65	24.40 ± 3.02	0.009 **
Timed Up and Go Test, seconds	6.08 ± 1.05	6.30 ± 0.75	7.33 ± 0.81	<0.001 ***
Berg Balance Scale	-	-	54.97 ± 1.43	-
Falling event, *n*	0	0	3	-

Data are number (%) or mean ± SD. * *p* < 0.05; ** *p* < 0.01; and *** *p* < 0.001.

**Table 2 sensors-24-07078-t002:** Analysis of balance board stance test with feet-apart stance.

J_balance_	Young	Middle-Aged	Older	*p*-Value	Post Hoc Analysis	
Waist	0.012 (0.006)	0.018 (0.017)	0.018 (0.035)	<0.001 ***	Y-M	0.0035 ^††^
					Y-O	<0.001 ^†††^
					M-O	0.4925
Chest	0.018 (0.004)	0.025 (0.030)	0.030 (0.038)	<0.001 ***	Y-M	0.0063 ^††^
					Y-O	<0.001 ^†††^
					M-O	0.0492
Right knee	0.010 (0.006)	0.019 (0.057)	0.022 (0.077)	<0.001 ***	Y-M	0.0014 ^††^
					Y-O	<0.001 ^†††^
					M-O	0.1289
Left knee	0.012 (0.005)	0.022 (0.054)	0.022 (0.081)	<0.001 ***	Y-M	0.0019 ^††^
					Y-O	<0.001 ^†††^
					M-O	0.1945
Right ankle	0.013 (0.008)	0.022 (0.051)	0.027 (0.069)	<0.001 ***	Y-M	<0.001 ^†††^
					Y-O	<0.001 ^†††^
					M-O	0.3473
Left ankle	0.014 (0.007)	0.025 (0.044)	0.026 (0.073)	<0.001 ***	Y-M	<0.001 ^†††^
					Y-O	<0.001 ^†††^
					M-O	0.7885
Foot	0.008 (0.004)	0.018 (0.018)	0.017 (0.035)	<0.001 ***	Y-M	0.001 ^††^
					Y-O	<0.001 ^†††^
					M-O	0.1731

Data are median (IQR). *** *p* < 0.001. Y-M: young vs. middle-aged; Y-O: young vs. older; M-O: middle-aged vs. older. Bonferroni correction was applied for multiple comparisons, adjusting the *p*-value threshold for significance to 0.05/3 = 0.0167. ^††^
*p* < 0.01; ^†††^
*p* < 0.001.

**Table 3 sensors-24-07078-t003:** Analysis of balance board stance test with feet-together stance.

J_balance_	Young	Middle-Aged	Older	*p*-Value	Post Hoc Analysis	
Waist	0.022 (0.016)	0.035 (0.023)	0.044 (0.067)	<0.001 ***	Y-M	0.0152 ^†^
					Y-O	<0.001 ^†††^
					M-O	0.0458
Chest	0.026 (0.010)	0.050 (0.025)	0.062 (0.086)	<0.001 ***	Y-M	<0.001 ^†††^
					Y-O	<0.001 ^†††^
					M-O	0.0333
Right knee	0.029 (0.024)	0.049 (0.044)	0.070 (0.122)	<0.001 ***	Y-M	0.0078 ^††^
					Y-O	<0.001 ^†††^
					M-O	0.1430
Left knee	0.030 (0.027)	0.051 (0.046)	0.073 (0.120)	<0.001 ***	Y-M	0.0115 ^†^
					Y-O	<0.001 ^†††^
					M-O	0.1559
Right ankle	0.032 (0.023)	0.046 (0.044)	0.059 (0.113)	0.001 **	Y-M	0.0167
					Y-O	<0.001 ^†††^
					M-O	0.159
Left ankle	0.033 (0.021)	0.052 (0.033)	0.062 (0.109)	<0.001 ***	Y-M	0.005 ^††^
					Y-O	<0.001 ^†††^
					M-O	0.2720
Right foot	0.018 (0.012)	0.027 (0.025)	0.041 (0.065)	<0.001 ***	Y-M	0.0239
					Y-O	<0.001 ^†††^
					M-O	0.067

Data are median (IQR). ** *p* < 0.01; and *** *p* < 0.001. Y-M: young vs. middle-aged; Y-O: young vs. older; M-O: middle-aged vs. older. Bonferroni correction was applied for multiple comparisons, adjusting the *p*-value threshold for significance to 0.05/3 = 0.0167. ^†^
*p* < 0.0167; ^††^
*p* < 0.01; ^†††^*p* < 0.001.

**Table 4 sensors-24-07078-t004:** Analysis of single-leg stance test with right foot stance.

J_balance_	Young	Middle-Aged	Older	*p*-Value	Post Hoc Analysis	
Waist	0.033 (0.015)	0.042 (0.019)	0.078 (0.060)	<0.001 ***	Y-M	0.0168
					Y-O	<0.001 ^†††^
					M-O	<0.001 ^†††^
Chest	0.040 (0.010)	0.052 (0.038)	0.091 (0.044)	<0.001 ***	Y-M	0.0166 ^†^
					Y-O	<0.001 ^†††^
					M-O	<0.001 ^†††^
Right wrist	0.042 (0.013)	0.051 (0.048)	0.116 (0.059)	<0.001 ***	Y-M	0.0065 ^††^
					Y-O	<0.001 ^†††^
					M-O	<0.001 ^†††^
Left wrist	0.042 (0.015)	0.050 (0.047)	0.114 (0.060)	<0.001 ***	Y-M	0.0093 ^††^
					Y-O	<0.001 ^†††^
					M-O	<0.001 ^†††^
Right ankle	0.135 (0.105)	0.160 (0.098)	0.210 (0.108)	<0.001 ***	Y-M	0.2221
					Y-O	<0.001 ^†††^
					M-O	0.0162 ^†^
Left ankle	0.044 (0.019)	0.060 (0.030)	0.103 (0.068)	<0.001 ***	Y-M	0.0217
					Y-O	<0.001 ^†††^
					M-O	<0.001 ^†††^
Foot	0.067 (0.028)	0.111 (0.064)	0.187 (0.095)	<0.001 ***	Y-M	0.0059 ^††^
					Y-O	<0.001 ^†††^
					M-O	<0.001 ^†††^
Head	0.043 (0.009)	0.061 (0.040)	0.094 (0.037)	<0.001 ***	Y-M	0.0132 ^†^
					Y-O	<0.001 ^†††^
					M-O	<0.001 ^†††^

Data are median (IQR). *** *p* < 0.001. Y-M: young vs. middle-aged; Y-O: young vs. older; M-O: middle-aged vs. older. Bonferroni correction was applied for multiple comparisons, adjusting the *p*-value threshold for significance to 0.05/3 = 0.0167. ^†^ *p* < 0.0167; ^††^ *p* < 0.01; ^†††^ *p* < 0.001.

**Table 5 sensors-24-07078-t005:** Analysis of single-leg stance test with left foot stance.

J_balance_	Young	Middle-Aged	Older	*p*-Value	Post Hoc Analysis	
Waist	0.033 (0.011)	0.045 (0.019)	0.079 (0.052)	<0.001 ***	Y-M	0.0240
					Y-O	<0.001 ^†††^
					M-O	<0.001 ^†††^
Chest	0.039 (0.013)	0.054 (0.035)	0.085 (0.037)	<0.001 ***	Y-M	0.0214
					Y-O	<0.001 ^†††^
					M-O	<0.001 ^†††^
Right wrist	0.041 (0.011)	0.059 (0.039)	0.101 (0.063)	<0.001 ***	Y-M	0.0313
					Y-O	<0.001 ^†††^
					M-O	<0.001 ^†††^
Left wrist	0.042 (0.015)	0.058 (0.040)	0.113 (0.069)	<0.001 ***	Y-M	0.0277
					Y-O	<0.001 ^†††^
					M-O	<0.001 ^†††^
Right ankle	0.045 (0.016)	0.061 (0.027)	0.092 (0.073)	<0.001 ***	Y-M	0.0406
					Y-O	<0.001 ^†††^
					M-O	<0.001 ^†††^
Left ankle	0.101 (0.063)	0.132 (0.126)	0.249 (0.088)	<0.001 ***	Y-M	0.0070 ^††^
					Y-O	<0.001 ^†††^
					M-O	0.0033 ^††^
Foot	0.066 (0.031)	0.113 (0.072)	0.182 (0.113)	<0.001 ***	Y-M	0.0103 ^†^
					Y-O	<0.001 ^†††^
					M-O	<0.001 ^†††^
Head	0.050 (0.017)	0.062 (0.031)	0.103 (0.046)	<0.001 ***	Y-M	0.0175
					Y-O	<0.001 ^†††^
					M-O	<0.001 ^†††^

Data are median (IQR). *** *p* < 0.001. Y-M: young vs. middle-aged; Y-O: young vs. older; M-O: middle-aged vs. older. Bonferroni correction was applied for multiple comparisons, adjusting the *p*-value threshold for significance to 0.05/3 = 0.0167. ^†^
*p* < 0.0167; ^††^
*p* < 0.01; ^†††^
*p* < 0.001.

## Data Availability

The dataset in this paper is available as follows: The IMU data of all subjects: http://gofile.me/4Zuw3/uD3ACh6t4, accessed on 23 August 2024.

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
