# Peer review of "Age-Related Influence on Static and Dynamic Balance Abilities: An Inertial Measurement Unit-Based Evaluationâ€"

_sensors, 2024, doi:10.3390/s24217078_

Round 1
Reviewer 1 Report
Comments and Suggestions for Authors
Dear authors
Congratulations on your valuable work. While your study offers an interesting exploration of age-related changes in static and dynamic balance using IMUs, I suggest emphasizing on the contributions of the study. You should clearly articulate how this study advances existing knowledge in the field. While the age-related decline in balance is extensively well-documented, the authors' approach using IMUs to differentiate the stages of balance decline between young, middle-aged, and older adults could be more clearly presented as a key contribution.
Despite the study presents valuable findings regarding age-related changes in balance, I would like to point out that the use of inertial measurement units (IMUs) to assess balance cannot be considered a novelty. The application of IMUs, particularly using angular velocity data, has been well-established for decades in both research and clinical settings for evaluating gait and postural stability. IMUs have long been recognized for their ability to provide objective and precise measurements of balance, making them a standard tool in the field.
Additionally, the research gap that motivated this study remains implicit. I suggest explicitly stating what gaps in the literature or limitations of previous balance assessment methods were identified and how this study seeks to address them.
I believe the discussion section is somewhat artificial and could be significantly improved. Currently, it reads more like a summary of the results alongside a collection of well-known facts. A strong discussion should critically compare the study’s findings with previous research, highlighting how these results confirm, contradict, or extend prior knowledge in the field. I encourage the authors to engage with the existing literature and provide a thorough comparison of their results with those of similar studies. Additionally, discussing potential reasons for any discrepancies or novel findings will enrich the discussion and demonstrate the broader impact of the study.
Author Response
Comments 1: Despite the study presents valuable findings regarding age-related changes in balance, I would like to point out that the use of inertial measurement units (IMUs) to assess balance cannot be considered a novelty. The application of IMUs, particularly using angular velocity data, has been well-established for decades in both research and clinical settings for evaluating gait and postural stability. IMUs have long been recognized for their ability to provide objective and precise measurements of balance, making them a standard tool in the field. Additionally, the research gap that motivated this study remains implicit. I suggest explicitly stating what gaps in the literature or limitations of previous balance assessment methods were identified and how this study seeks to address them. |
Response 1: We thank the reviewer’s suggestion. Although several studies have used inertial measurement units (IMUs) to assess gait and postural stability, only a few have investigated age-related changes in balance. Among these studies, most focused on either static or dynamic balance, with IMUs primarily placed on the trunk to evaluate trunk stability. Additionally, many studies limited their analysis to parameters derived from accelerometer. To address these gaps, we present a more comprehensive study that utilizes a greater number of IMUs to measure participants' kinematic data across various body parts, including the head, chest, waist, knees, ankles, and feet, in order to assess both static and dynamic balance abilities. We also evaluated multiple indices, incorporating various combinations of angular and linear velocities. Our results showed that in the dynamic balance test, both middle-aged and older adults had significantly higher average absolute angular velocities across all body segments compared to young adults. However, there was no significant difference between the middle-aged and older groups, suggesting a greater increase in postural sway when transitioning from young adulthood to middle age, with a smaller increase as one ages further. In contrast, the static balance test revealed significant differences in the single-leg stance between young and older adults, as well as between middle-aged and older adults, but not between young and middle-aged groups. This indicates that postural sway increases more notably from middle age to older age during a single-leg stance. These results suggest that dynamic balance (balance board) declines more rapidly than static balance (single-leg stance). Additionally, significant differences between young and middle-aged groups were found in only three of the six body segments (chest, wrist, and foot) during the single-leg stance test. These findings underscore the importance of considering multiple body segments and balance assessment methods when investigating age-related changes in postural control. To our knowledge, this observation has not been highlighted in previous studies. Action 1: We have revised and highlighted the manuscript to emphasize this point in the “Introduction” section (Page 2, lines 59-74 and Page 3, lines 103-123) and in the “Discussion” section (Page 17-18, lines 431-493). |
Comments 2: I believe the discussion section is somewhat artificial and could be significantly improved. Currently, it reads more like a summary of the results alongside a collection of well-known facts. A strong discussion should critically compare the study’s findings with previous research, highlighting how these results confirm, contradict, or extend prior knowledge in the field. I encourage the authors to engage with the existing literature and provide a thorough comparison of their results with those of similar studies. Additionally, discussing potential reasons for any discrepancies or novel findings will enrich the discussion and demonstrate the broader impact of the study. |
Response 2: We appreciate this feedback and have revised the “Discussion” section to provide a more critical comparison of our study’s findings with previous research. Specifically, we have engaged with relevant literature to highlight how our results confirm, contradict, or extend existing knowledge in the field. Additionally, we have explored potential reasons for any observed discrepancies or novel findings, thereby providing a deeper understanding of the study's broader impact. We hope these improvements address your concerns and enrich the overall discussion. Actions 2: We have revised and highlighted the “Discussion” section of the manuscript to emphasize this point. (Page 17-18, lines 431-493) |

Reviewer 2 Report
Comments and Suggestions for Authors
The manuscript entitled “Age-related influence on static and dynamic balance abilities: an inertial measurement unit-based evaluation” presents comparisons of balance performance across young, middle, and older aged adults standing on a balance board and in single leg stance. While the manuscript clearly demonstrates a large amount of effort and is generally well written, there are key issues brought out in this review.
My biggest concern is that the authors do not summarize the body of literature to identify a gap in knowledge, nor do they present a research question or hypothesis. There have been tons of studies focused on 1) investigating balance using IMUs and 2) investigating age-related changes to balance control. It is unclear what new information is provided by this paper.
Related to this, I am concerned with the statistical analysis of this paper. With no clear question, the authors performed a large number of statistical comparisons and failed to correct their level of significance.
Additionally, the outcome measures assessed in this study do not align with prior literature using IMUs to assess balance nor do they have strong theoretical basis for being an appropriate measure of balance performance.
Introduction
Lines 40-45 provide make multiple statements with no reference which need one. Additionally, the one reference provided seems to investigate effects of age on balance control, which does not support claims made that balance is associated with risk of injurious falls or that falls lead to injury and other complications.
Lines 46-58 describe why balance ability decreases with age. However, it doesn’t describe balance changes relevant to this study. Have studies compared ability to maintain balance on a balance board between younger and older adults? Have they compared single leg balance between young/old? What specifically is unknown?
Lines 69-80 describe various instrumentation used to assess balance, leaving off with only three studies which used IMU to assess balance in yoga dancers and Parkinson’s disease. First off, there are a ton of studies which have used IMUs to assess balance in older adults and various patient populations. Those would be much more pertinent here. Second, just saying what the study did in one sentence is not enough. You need to establish the limitations of prior research to make it very clear what the gap in literature is that your study hopes to fill.
Lines 82-97 describe motivation for the study then summarize the methods. Nowhere in here is a specific research question or testable hypothesis.
Methods
It is not necessary to provide the specifications for a commercially available product like the Opal sensors. These sensors are not new, they have been used for published research for over 15 years.
This specific balance board seems novel. However it is not introduced or explained why this balance board. What was the shortcomings of traditional balance boards (i.e., wobble boards) or foam pads used in prior research? This should be the focus of your entire methods section – reviewing the prior literature in this area.
What was the stiffness of those cables on the balance board? I’m trying to picture how hard of a task this was relative to traditional balance boards.
Lines 152-158 – I have never seen balance data from IMUs analyzed like this. You basically just summed every data point for each segment across the whole trial. What does this tell you? Please refer to the IMU balance literature and analyze the data using accepted methods.
Results
Oh, I see now that you did analyze every sensor separately. That was not clear.
Lines 192 – 206 seem to represent information repeating from the table. Just summarize the table here. Saying the percent which average angular velocity for one segment was decreased relative to older adults doesn’t hold much weight without standard deviation and statistical comparisons provided in the table.
There are way to many statistical tests performed in this study. I suggest that, after you develop a hypothesis, you choose one primary outcome measures and a handful of secondary outcomes then only run and present statistical measures which specifically test that hypothesis.
Author Response
Comments 1: My biggest concern is that the authors do not summarize the body of literature to identify a gap in knowledge, nor do they present a research question or hypothesis. There have been tons of studies focused on 1) investigating balance using IMUs and 2) investigating age-related changes to balance control. It is unclear what new information is provided by this paper. |
Response 1: We thank the reviewer for the valuable comments. As our reply to Reviewer 1, although several studies have used IMUs to assess gait and postural stability, only a few have investigated age-related changes in balance. Among these studies, most focused on either static or dynamic balance, with IMUs primarily placed on the lower trunk to evaluate trunk stability. Additionally, many studies limited their analysis to linear accelerations. This leaves several fundamental questions: Do age-related changes in balance differ between static and dynamic conditions? Can a more comprehensive set of IMUs, placed on multiple body segments, detect more age-related differences compared to assessments using a single IMU? To address these gaps, we conducted a study that employs a larger number of IMUs to collect kinematic data from various body parts, including the chest, waist, knees, ankles, and feet, to assess both static and dynamic balance. Additionally, we evaluated two indices, considering average absolute angular velocity and average absolute linear accelerations. Action 1: We have summarized the existing body of literature to identify a gap in knowledge in the “Introduction” section (Page 2, lines 59-74) and in the “Discussion” section (Page 17-18, 431-493). Additionally, we present key research questions in the “Introduction” section, including: Do age-related changes in balance differ between static and dynamic conditions? Can a more comprehensive set of IMUs, placed on multiple body segments, detect more age-related differences compared to assessments using a single IMU? (Page 3, lines 116-119) |
Comments 2: Related to this, I am concerned with the statistical analysis of this paper. With no clear question, the authors performed a large number of statistical comparisons and failed to correct their level of significance. |
Response 2: As one of our study aim is to determine if a multi-IMU approach can detect more age-related differences than a single-IMU approach, we conducted statistical comparisons among three age groups across various body segments. We selected the balance index at the waist as the primary outcome for both static and dynamic balance, aligning with the focus on the waist as the center of mass in most previous studies. We further collect kinematic data from various body parts separately, including the chest, knees, ankles, and feet, to evaluate whether certain IMU placements are more effective in distinguishing between different age groups. Our results revealed significant differences between the young and middle-aged groups in only three of the six body segments: the chest, wrist, and foot during the single leg-stance test. These findings highlight the importance of considering multiple body segments and balance assessment methods when investigating age-related changes in postural control. While the waist may be a valuable marker for overall balance, additional measurements from other body parts may provide more nuanced insights into the specific mechanisms underlying balance decline with age. Action 2: We have revised and highlighted the “Introduction” section (Page 3, lines 116-119) and “Material and method” (Pages 8, lines 259-260) of the manuscript to emphasize this point. |
Comments 3: Additionally, the outcome measures assessed in this study do not align with prior literature using IMUs to assess balance nor do they have strong theoretical basis for being an appropriate measure of balance performance. |
Response 3: In our previous work, we defined a balance index called average absolute angular velocity, which successfully differentiated the balance abilities between Latin dancers, yoga instructors, and age-matched adults (see refs [28] [29] in revised manuscript). For the pilot study (see supplemental data), we initially tested two indices, including average absolute angular velocity ( ) and average absolute linear accelerations ( ). The results indicated that average absolute angular velocity ( ) was the more effective index for assessing balance ability. Therefore, in this study, we employed average absolute angular velocities ( ) to distinguish between static and dynamic balance abilities across different age groups. Action 3: We have revised and highlighted the “Material and methods” section of the manuscript to emphasize this point (Page 8, lines 259-265). |
Comments 4: Introduction Lines 40-45 provide make multiple statements with no reference which need one. Additionally, the one reference provided seems to investigate effects of age on balance control, which does not support claims made that balance is associated with risk of injurious falls or that falls lead to injury and other complications. |
Response 4: We thank the reviewer for the feedback and comments. Action 4: We have added the references to the “Introduction” section accordingly (Page 1, lines 40-45). |
Comments 5: Lines 46-58 describe why balance ability decreases with age. However, it doesn’t describe balance changes relevant to this study. Have studies compared ability to maintain balance on a balance board between younger and older adults? Have they compared single leg balance between young/old? What specifically is unknown? |
Response 5: We thank the reviewer’s feedback. Previous studies investigating aged-related balance decline during one-leg stance have often relied on traditional methods, such as measuring stance time or using force plates. Similarly, research on age-related deterioration of dynamic balance using balance boards has primarily utilized reaction time measurements or force plates. The above approach primarily focused on overall stability metrics derived from a central point of measurement and may fail to capture the intricate contributions of various body segments to postural control. A notable research gap exists in the use of more advanced methodologies, such as deploying multiple IMUs across different body segments, which could yield a more detailed and comprehensive analysis of balance dynamics. Furthermore, these studies have often not included a sufficiently diverse range of age groups, typically concentrating on either young or older adults while overlooking middle-aged individuals. This results in a limited understanding of how balance evolves throughout the aging process. Therefore, our study aimed to address these gaps through the application of IMUs and a more inclusive study population. Action 5: we have added the information to the “Introduction” section accordingly (Page 2, lines 59-74). |
Comments 6: Lines 69-80 describe various instrumentation used to assess balance, leaving off with only three studies which used IMU to assess balance in yoga dancers and Parkinson’s disease. First off, there are a ton of studies which have used IMUs to assess balance in older adults and various patient populations. Those would be much more pertinent here. Second, just saying what the study did in one sentence is not enough. You need to establish the limitations of prior research to make it very clear what the gap in literature is that your study hopes to fill. |
Response 6: We thank the reviewer’s feedback. Although numerous studies have utilized IMUs to assess gait and postural stability, few have specifically examined age-related changes in balance. Most of these studies focused solely on either static or dynamic balance, with IMUs mainly positioned on the lower trunk to assess trunk stability, and many limited their analyses to accelerations. Using a single IMU limits kinematic data to one body segment, which restricts the assessment of coordination between different body parts. This approach may miss crucial segmental movements needed for balance, as effective balance often involves coordinated actions across multiple joints. It can also reduce sensitivity to subtle balance impairments, especially in those with age-related or neurological conditions. Additionally, assuming that trunk movement represents overall body balance can be inaccurate, particularly during dynamic activities. Moreover, evaluating only static or dynamic balance abilities in age-related balance decline can provide a limited understanding of the underlying mechanisms and risk factors for falls. We have summarized the existing body of literature to identify a gap in knowledge Action 6: We have addressed the limitations of prior research and identify a gap in knowledge in the “Introduction” section accordingly (Page 2, lines 59-74). |
Comments 7: Lines 82-97 describe motivation for the study then summarize the methods. Nowhere in here is a specific research question or testable hypothesis. |
Response 7: We thank the reviewer for the valuable feedback.
Action 7: We have presented key research questions in the “Introduction” section, including: Do age-related changes in balance differ between static and dynamic conditions? Can a more comprehensive set of IMUs, placed on multiple body segments, detect more age-related differences compared to assessments using a single IMU (Page 3, lines 116-119)? |
Comments 8: Methods |
Response 8: We thank the reviewer for the comments and have removed the specific details of the Opal IMU system listed in Table 1. Action 8: We have removed the specific details of the Opal IMU system listed in Table 1. (Page 4). |
Comments 9: This specific balance board seems novel. However, it is not introduced or explained why this balance board. What was the shortcomings of traditional balance boards (i.e., wobble boards) or foam pads used in prior research? This should be the focus of your entire methods section – reviewing the prior literature in this area. What was the stiffness of those cables on the balance board? I’m trying to picture how hard of a task this was relative to traditional balance boards. |
Response 9: We appreciate the reviewer’s comments and have incorporated a summary of relevant articles on traditional balance boards, along with an explanation of why we chose to use the balance board. Additionally, we have included a more detailed description of the balance board. Action 9: We have modified the revised manuscript in the “Material and methods” section (Page 4, lines 154-179). Figure R2 was added. |
Comments 10: Lines 152-158 – I have never seen balance data from IMUs analyzed like this. You basically just summed every data point for each segment across the whole trial. What does this tell you? Please refer to the IMU balance literature and analyze the data using accepted methods. |
Response 10: As one of our study aim is to determine if a multi-IMU approach can detect more age-related differences than a single-IMU approach, we conducted statistical comparisons among three age groups across various body segments. We selected the balance index at the waist as the primary outcome for both static and dynamic balance, aligning with the focus on the waist as the center of mass in most previous studies. We further collect kinematic data from various body parts separately, including the chest, knees, ankles, and feet, to evaluate whether certain IMU placements are more effective in distinguishing between different age groups. Our results revealed significant differences between the young and middle-aged groups in only three of the six body segments: the chest, wrist, and foot during the single leg-stance test. These findings highlight the importance of considering multiple body segments and balance assessment methods when investigating age-related changes in postural control. While the waist may be a valuable marker for overall balance, additional measurements from other body parts may provide more nuanced insights into the specific mechanisms underlying balance decline with age. Traditionally, the most commonly used parameters are the RMS of linear acceleration in the AP and/or ML directions. In previous studies, the range of angular velocity in the pitch and roll directions has also been used for assessment. Our prior experience suggests that angular velocity data tends to be more sensitive. Furthermore, we recognize that during balance tasks, angular velocity in all three directions—pitch, roll, and yaw—may contribute to the movement. Based on this, we defined the average absolute angular velocity, calculated by summing the squares of the angular velocities in all three directions within a period of time, effectively eliminating directional bias and yielding the total angular velocity within that time. Since each individual stabilizes differently during testing and experiences varying degrees of sway over time, we then averaged the absolute angular velocities to obtain the mean absolute angular velocity. Another reason why angular velocity may be more sensitive is due to the nature of our tests, which involve no movement of the base of support. When participants exhibit smaller movements, linear acceleration may be less detectable; in contrast, angular velocity becomes more sensitive in such conditions. Although previous studies focused predominantly on angular velocity in the pitch and roll directions, we believe that movements in all three directions should be considered, as all are likely involved during balance tasks. Action 10: We have revised and highlighted the “Material and methods” (Page 6, lines 213-221) and “Discussion” (Page 17-18, lines 463-493) section of the manuscript to emphasize this point. |
Comments 11: Results Oh, I see now that you did analyze every sensor separately. That was not clear. |
Response 11: We thank the reviewer for the feedback. Action 11: We have revised and highlighted the “Introduction” (Pages 3, lines 119-123) section of the manuscript to emphasize that we analyzed every sensor separately. |
Comments 12: Lines 192 – 206 seem to represent information repeating from the table. Just summarize the table here. Saying the percent which average angular velocity for one segment was decreased relative to older adults doesn’t hold much weight without standard deviation and statistical comparisons provided in the table. |
Response 12: We thank the reviewer for the feedback. Action 12: We have removed the percentage for the average angular velocity that decreased relative to older adults in the “Results” Section and simply summarized the table (Pages 8, lines 266-275),(Page 8-9, lines 283-292), (Page 11-12, lines 318-324), (Page 13, lines 335-339). |
Comments 13: There are way too many statistical tests performed in this study. I suggest that, after you develop a hypothesis, you choose one primary outcome measures and a handful of secondary outcomes then only run and present statistical measures which specifically test that hypothesis. |
Response 13: As one of our study aim is to determine if a multi-IMU approach can detect more age-related differences than a single-IMU approach, we conducted statistical comparisons among three age groups across various body segments. We selected the balance index at the waist as the primary outcome for both static and dynamic balance, aligning with the focus on the waist as the center of mass in most previous studies. We further collect kinematic data from various body parts separately, including the chest, knees, ankles, and feet, to evaluate whether certain IMU placements are more effective in distinguishing between different age groups. |

Round 2
Reviewer 1 Report
Comments and Suggestions for Authors
Dear authors, I have no further comments on your paper, as all my suggestions have been addressed. I believe it is ready for acceptance in its current form.
Author Response
Dear Reviewer,
Thank you very much for your thorough review and constructive feedback throughout this process. We appreciate the time and effort you invested in helping us enhance the quality of our paper. We are pleased to hear that our revisions have met your expectations, and we are grateful for your recommendation for acceptance.
Reviewer 2 Report
Comments and Suggestions for Authors
The authors of the manuscript entitled “Age-related influence on static and dynamic balance abilities: an inertial measurement unit-based evaluation” did a nice job addressing my concerns. There are a few more questions raised in this review below.
The authors seem to have only added information to the introduction section, rather than revise the sections content. I appreciate the authors demonstrating the gap in that only a few prior studies have used IMUs to assess age related changes in balance and have only assessed static or dynamic balance. The authors should go into details about a few of these studies to really show why their question is novel and why it is important. By details I mean what the study did and what they found, with some limitations. I just did a literature search and found tons of studies that mention IMU balance and age in their title.
The authors list specific research questions, but not in a way which is typical in the scientific literature. Additionally, there are no hypotheses listed for these research questions. Each question should have a separate hypothesis with literature-backed justification. Usually, papers say something like: “The first purpose of this study was to determine the effects of X on Y and Z. Based on prior literature (REF), we hypothesized that X would increase Y but decrease Z”.
Along this line, the authors research question “Do age related changes in balance differ between static and dynamic conditions?” is not adequately addressed through the statistical analyses presented because the authors are not statistically comparing the differentiating capabilities of static and dynamic balance. This research question is set up to be answered using a two-way ANOVA model (young vs old as between subjects and static vs dynamic as within subject).
The last paragraph of the introduction section is repetitive with the new purpose/hypotheses paragraph and should be omitted (line 125-135).
It is unclear why the authors introduced new content on mathematically modeling the balance board and deriving control equations. This content does not address the research questions and was not used in the analysis. It does help describe the balance board test, but that can be done with words. It is nice to see that this balance board is commercially available and used in clinical populations.
The correlation plots still list J-balance. Is that what you are calling J-w now? What different information is the correlation analysis telling you relative to the one way anova?
In general, this paper is really long. I think the authors should focus on being more concise to hold their audience’s attention and communicate the important information from the work.
Author Response
Comments 1: The authors seem to have only added information to the introduction section, rather than revise the sections content. I appreciate the authors demonstrating the gap in that only a few prior studies have used IMUs to assess age related changes in balance and have only assessed static or dynamic balance. The authors should go into details about a few of these studies to really show why their question is novel and why it is important. By details I mean what the study did and what they found, with some limitations. I just did a literature search and found tons of studies that mention IMU balance and age in their title. |
Response 1: Thank you for your valuable feedback. We agree with your suggestions. Action 1: We have now added detailed information regarding several studies in the revised manuscript to highlight the novelty and importance of our research question (Page 2, lines 82-91, 118-120). |
Comments 2: The authors list specific research questions, but not in a way which is typical in the scientific literature. Additionally, there are no hypotheses listed for these research questions. Each question should have a separate hypothesis with literature-backed justification. Usually, papers say something like: “The first purpose of this study was to determine the effects of X on Y and Z. Based on prior literature (REF), we hypothesized that X would increase Y but decrease Z”. Along this line, the authors research question “Do age related changes in balance differ between static and dynamic conditions?” is not adequately addressed through the statistical analyses presented because the authors are not statistically comparing the differentiating capabilities of static and dynamic balance. This research question is set up to be answered using a two-way ANOVA model (young vs old as between subjects and static vs dynamic as within subject). |
Response 2: We appreciate the feedback and have revised our hypotheses and research questions accordingly. Actions 2: The introduction section was rewritten to address this problem (Page 4, lines 178-193). |
Comments 3: The last paragraph of the introduction section is repetitive with the new purpose/hypotheses paragraph and should be omitted (line 125-135). |
Response 3: We appreciate the comments and had revised the manuscript accordingly. Actions 3: The last paragraph has been removed and the introduction section has been rewritten. |
Comments 4: It is unclear why the authors introduced new content on mathematically modeling the balance board and deriving control equations. This content does not address the research questions and was not used in the analysis. It does help describe the balance board test, but that can be done with words. It is nice to see that this balance board is commercially available and used in clinical populations. |
Response 4: We appreciate the feedback and have revised the manuscript accordingly. Actions 4: We had deleted the mathematically modeling of the balance board and control equations, retaining only the descriptive text for the balance board test and Figure R2(a). |
Comments 5: The correlation plots still list J-balance. Is that what you are calling J-w now? What different information is the correlation analysis telling you relative to the one way anova? |
Response 5: Yes, we have changed the J-balance into J-w to emphasize the angular velocity (omega, ) and to differentiate from J-a (acceleration, which we have added to the manuscript). The correlation analysis provides additional insights into the relationship between age and J-w by highlighting trends across different body sites. This approach allows us to see patterns that the one-way ANOVA does not capture, thereby verifying our conclusions using complementary statistical methods. Actions 5: We have changed all the J-balance into J-w in the correlation plots. |
Comments 6: In general, this paper is really long. I think the authors should focus on being more concise to hold their audience’s attention and communicate the important information from the work. |
Response 6: Thank you for the feedback. We have revised the introduction section to enhance conciseness and have removed redundant content in the discussion section (Page 18, lines 478-483) to improve clarity and maintain reader engagement. |
